# Untangling irrigation effects on maize water and heat stress alleviation using satellite data

Peng Zhu[1*], Jennifer Burney[1]
[1]School of Global Policy and Strategy, University of California, San Diego, CA USA
*Correspondence to*: Peng Zhu (zhuyp678@gmail.com)

Abstract**.** Irrigation has important implications for sustaining global food production, enabling crop water demand to be met even under dry conditions. Added water also cools crop plants through transpiration; irrigation might thus play an important role in a warmer climate by simultaneously moderating water and high temperature stresses. Here we use satellite-derived evapotranspiration estimates, land surface temperature (LST) measurements, and crop phenological stage information from Nebraska maize to quantify how irrigation relieves both water and temperature stresses. Our study shows that, unlike air temperature metrics, satellite-derived LST detects significant irrigation-induced cooling effect, especially during the grain filling period (GFP) of crop growth. This cooling is likely to extend the maize growing season, especially for GFP, likely due to the stronger temperature sensitivity of phenological development during this stage. The analysis also suggests that irrigation not only reduces water and temperature stress but also weakens the response of yield to these stresses. Specifically, temperature stress is significantly weakened for reproductive processes in irrigated crops. The attribution analysis further suggests that water and high temperature stress alleviation contributes to 65% and 35% of yield benefit, respectively. Our study underlines the relative importance of high temperature stress alleviation in yield improvement and the necessity of simulating crop surface temperature to better quantify heat stress effects in crop yield models. Finally, untangling irrigation effects on both heat and water stress mitigation has important implications for designing agricultural adaptation strategies under climate change.

**Keywords: Irrigation, Evaporative cooling, MODIS LST, High temperature stress, Water stress, Maize**

## 1. Introduction

Irrigation -- a large component of freshwater consumption sourced from water diversion from streams and groundwater (Wallace, 2000, Howell, 2001) -- allows crops to grow in environments that do not receive sufficient rainfall, and buffers agricultural production from climate variability and extremes. Irrigated agriculture plays an outsized role in global crop production and food security: irrigated lands account for 17% of total cropped area, yet they provide 40% of global cereals (Rosegrant et al 2002, Siebert and Döll 2010). Meeting the rising food demands of a growing global population will require either increasing crop productivity and/or expansion of cropped areas; both strategies are daunting under projected climate change. Cropland expansion may be in marginal areas that require irrigation even in the present climate (Bruinsma 2009); increasing temperatures will drive higher atmospheric vapor pressure deficits (VPD) and raise crop water demand and crop water losses. This increasing water demand poses a water ceiling for crop growth and might necessitate irrigation application over present rainfed areas to increase or even maintain yields (DeLucia et al., 2019).

However, the provision of additional irrigation water modifies both the land surface water and energy budgets. Additional water can result in an evaporative cooling effect, which may be beneficial for crop growth indirectly through lowering the frequency of extreme heat stress (Butler et al., 2018). Especially considering the future warmer climate, high temperature stress will be more prevalent (Russo et al., 2014) and might result in more severe yield losses than water stress (Zhu et al., 2019) due to reduced photosynthesis, pollen sterility, and accelerated crop senescence in major cereals (Rezaei et al., 2015b; Rattalino Edreira et al., 2011; Ruiz-Vera et al., 2018), therefore, a better understanding of irrigation effect on high temperature stress alleviation will be important for agricultural management practices. More broadly, understanding how irrigation can or should contribute to a portfolio of agricultural adaptation strategies thus requires improved understanding of its relative roles in mitigating both water and heat stresses.

Climate models and meteorological data have been used to investigate how historical expansion of irrigation at global and regional scales has influenced the climate

Hydrology and Earth System Sciences
EGU
Discussions

system, including surface cooling and precipitation variation (Kang and Eltahir, 2019; Thiery et al., 2017; Bonfils and Lobell, 2007; Sacks et al., 2009). However, many crop models still use air temperature rather than canopy temperature to estimate heat stress; this may overestimate heat stress effect in irrigated cropland (Siebert et al., 2017), since canopy temperature can deviate significantly from air temperature depending on the crop moisture conditions (Siebert et al., 2014). Recently, a comparison of crop model simulated canopy temperature suggests that most crop models lack a sufficient ability to reproduce the field-measured canopy temperature, even for models with a good performance in grain yield simulation (Webber et al., 2017).

Alternatively, satellite-derived land surface temperature (LST) has been used to directly quantify regional scale surface warming or cooling effects resulting from surface energy budget changes due to changes in land cover and land management (Loarie et al., 2011; Tomlinson et al., 2012; Peng et al., 2014). Importantly, yield prediction model comparisons suggest that replacing air temperature with MODIS LST can improve yield predictions because LST accounts for both evaporative cooling and water stress (Li et al., 2019). Satellite data also provide the observational evidence to constrain model performance or directly retrieve crop growth status information. For example, satellite derived soil moisture had been used to characterize irrigation pattern and improve irrigation amount estimation (Felfelani et al., 2018; Lawston et al., 2017; Jalilvand et al., 2019; Zaussinger et al., 2019). Therefore, integrating satellite products have the potential to improve our understanding of how irrigation and climate change impact crop yield and thus provide guides for farmers to make the optimal decisions.

In this study, we focus on Nebraska, the third largest maize producer in the United States. Multi-year mean climate data shows that conditions are drier in western areas and warmer in southern areas (Figure 1a and b). Importantly, Nebraska features a mixture of irrigated and rainfed maize that facilitates comparison (more than half (56%) of the Nebraska maize cropland is irrigated with more irrigated maize in the western area (Figure 1c), according to the United States Department of Agriculture (USDA, 2018a)). County yield data from the USDA shows that interannual fluctuations in rainfed maize yield are much larger than for irrigated maize (Figure

1b). Although irrigated yields are higher, rainfed maize yields have grown faster than irrigated (3.9% per year versus 1.0% per year) over the study period (2003-2016) (Figure 1b), one of the possible reasons is that breeding technology progress has improved the drought tolerance of maize hybrids (Messina et al., 2010).

As noted above, irrigation potentially benefits crop yields by moderating both water and high temperature stress. Here we use satellite-derived LST and satellite-derived water stress metrics to statistically tease apart the contributions of irrigation to water and heat stress alleviation, separately. We: (1) evaluate the difference in temperature and moisture conditions over irrigated and rainfed maize croplands; (2) explore how irrigation mitigates water and high temperature stresses using panel statistical models; (3) quantify the relative contributions of irrigation-induced water and high temperature stress alleviation to yield improvements; and (4) explore whether current crop models can reproduce the observed irrigation benefits on maize growth status.

## 2. Materials and Methods

We first describe the data used, followed by a brief description of statistical methodology.

### 2.1 Satellite products to identify irrigated and non-irrigated maize areas

We used the United States Department of Agriculture's Cropland Data Layer (CDL) to identify maize croplands for each year in the study period 2003-2016 (USDA, 2018b). The irrigation distribution map across Nebraska was obtained from a previous study that used Landsat-derived plant greenness and moisture information to create a continuous annual irrigation map across U.S. Northern High Plains (Deines et al., 2017). The irrigation map showed a very high accuracy (92 to 100%) when validated with randomly generated test points and also highly correlated with county statistics ($R^2$ = 0.88–0.96) (Deines et al., 2017). Both the CDL and irrigation map are at 30m resolution. We first projected them to MODIS sinusoidal projection and then aggregated them to 1km resolution to align with MODIS ET and LST products. Then, pixels containing more than 60% maize and an irrigation fraction >60% were labeled as irrigated maize while pixels with >60% maize and <10% irrigation fraction were labeled as rainfed maize croplands. As always, threshold selection involves a tradeoff between mixing samples and retaining as many samples as possible. Our choices of

<10% as the threshold for rainfed maize and 60% to define irrigated maize represented the best optimization in our sample, as we found that more stringent threshold had a very small effect on LST differences between irrigated and rainfed maize at county level but resulted in significant data omission (more details in supplementary Figure 1-2).

## 2.2 Maize phenology information

Maize growth stage information derived in a previous study was used to assess the influence of irrigation on maize growth during different growth stages (Zhu et al., 2018). Stage information including emergence date, silking date, and maturity date, was derived with MODIS WDRVI (Wide Dynamic Range Vegetation Index, 8-day and 250m resolution) based on a hybrid method combining shape model fitting (SMF) and threshold-based analysis. Then we defined vegetative period (VP) as period from emergence date to silking date, grain filling period (GFP) as period from silking date to maturity date and growing season (GS) as period from emergence date to maturity date. Details can be found in our previous studies (Zhu et al., 2018). WDRVI was used due to its higher sensitivity to changes at high biomass than other vegetation indices (Gitelson et al., 2004) and was estimated with the following equation:

$$NDVI = (\rho_{NIR} - \rho_{red})/(\rho_{NIR} + \rho_{red}) \tag{1}$$

$$\text{WDRVI}=100 * \frac{[(\alpha-1)+(\alpha+1)\times NDVI]}{[(\alpha+1)+(\alpha-1)\times NDVI]} \tag{2}$$

where $\rho_{red}$ and $\rho_{NIR}$ were the MODIS surface reflectance in the red and NIR bands, respectively. To minimize the effects of aerosols, we used the 8-day composite products in MOD09Q1 and MYD09Q1 and quality-filtered the reflectance data using the band quality control flags. Only data passing the highest quality control were retained (Zhu et al., 2018). The scaling factor, α=0.1, was adopted based on a previous study to degrade the fraction of the NIR reflectance at moderate-to-high green vegetation and best linearly capture the maize green leaf area index (LAI) (Guindin-Garcia *et al.*, 2012).

## **2.3 Temperature exposure during maize growth**

We used daily 1-km spatial resolution MODIS Aqua LST (MYD11A1) data to characterize the crop surface temperature; since its overpassing time is at 1:30 and 13:30, it is closer to the times of daily minimum and maximum temperature than the

MODIS Terra LST (Wan et al., 2008) and is therefore better for characterizing crop surface temperature stress (Johnson 2016; Li et al., 2019). For quality control, pixels with an LST error >3 degree were filtered out based on the corresponding MODIS LST quality assurance layers. Missing values (less than 3%) were interpolated with robust spline function (Teuling et al., 2010). Aqua LST data are available after July 2002; we thus restricted our study to the period 2003-2016. For comparison, we also obtained minimum and maximum daily surface air temperature (Tmin and Tmax) at 1-km resolution from Daymet version 3 (Thornton et al., 2018). For both MODIS LST and air temperature, we calculated integrated crop heat exposure -- the growing degree days (GDD) and extreme degree days (EDD) -- with the following equations:

$$GDD_8^{30} = \sum_{t=1}^{N} DD_t,\ DD_t = \begin{cases} 0,\ when\, T < 8°\text{C} \\ T-8,\ when\, 8°\text{C} \le T < 30°\text{C} \\ 22,\ when\, T \ge 30°\text{C} \end{cases} \quad (3)$$

$$EDD_{30}^{\infty} = \sum_{t=1}^{N} DD_t,\ DD_t = \begin{cases} 0,\ when\, T < 30°\text{C} \\ T-30,\ when\, T \ge 30°\text{C} \end{cases} \quad (4)$$

Here temperature (*T*) could be either air temperature or LST and had been interpolated from daily to hourly values with sine function (Tack *et al.*, 2017). *t* represents the hourly time step, *N* is the total number of hours in a specified growing period (either the entire growing season, or a specific phenological growth phase, as defined below).

### 2.4 Maize Water Stress

Water stress during maize growth was characterized by the ratio of evapotranspiration (ET) to potential evapotranspiration (PET), as used in previous study (Mu et al., 2013). MODIS product (MYD16A2) provided both ET and PET from 2003 to 2016 and showed good performance for natural vegetation (Mu et al., 2011), however, our comparison using flux tower observed ET at an irrigated maize site at Nebraska suggested that ET at the irrigated maize was significantly underestimated by MODIS ET (Supplementary Figure 3). Therefore, we used another ET product (SSEBop ET) to replace MODIS ET. SSEBop ET was also estimated with MODIS products (Senay et al., 2013), like LST, vegetation index, and albedo as input variables, but used a revised algorithm including predefined boundary conditions for hot and cold reference pixels (Senay et al., 2013) and showed better performance than MODIS ET (Velpuri

et al., 2013), which was confirmed when we compared it with flux tower observed ET at an irrigated maize site (Supplementary Figure 4). The comparison of MODIS PET and flux tower estimated PET shows MODIS PET has satisfactory performance (Supplementary Figure 5). Since MODIS PET from MYD16A2 has a spatial resolution of 500 m with 8-day temporal resolution, while SSEBop ET has 1km spatial resolution with daily time step, we reconciled the two datasets to 1km spatial resolution and 8-day temporal resolution. Then ET, PET and ET/PET were averaged over time to get mean ET, PET and ET/PET during VP, GFP and GS with satellite derived phenology to characterize water status during maize growth.

### 2.5 Crop model simulation results

We compared the results of our statistical analysis with four gridded crop models. Simulation results from pAPSIM, pDSSAT, LPJ-GUESS, CLM-crop for both rainfed and irrigated maize across Nebraska were obtained from Agricultural Model Intercomparison and Improvement Project (AgMIP) (Rosenzweig *et al.*, 2013) and Inter-Sectoral Impact Model Intercomparison Project 1 (ISIMIP1) (Warszawski *et al.*, 2014). The four models were driven by the same climate forcing dataset (AgMERRA) and run at a spatial resolution of 0.5 arc-degree longitude and latitude. All simulations were conducted for purely rainfed and near-perfectly irrigated conditions. These models simulated maize yield, total biomass, ET and growing stage information (planting date, flowering date and maturity date). Planting date occurs on the first day following the prescribed sowing date in which soil temperature is at least 2 degrees above the 8 °C base temperature. Harvest occurs once the specified heat units are reached. Heat units to maturity were calibrated from the prescribed crop calendar data (Elliott et al., 2015). Crop model simulation was evaluated by calculating the Pearson correlation between simulated yields in the baseline simulations and detrended historical yields for each country from the Food and Agriculture Organization. Management scenario 'harmnon' was selected, meaning the simulation using harmonized fertilizer inputs and assumptions on growing seasons. More details on the simulation protocol can be found in Elliott et al. (2015) and Mueller et al. (2019). We used this model comparison project outputs to shed light on how well crop models had simulated the irrigation benefits we identified in different phases of crop growth.

**2.6 Method**

We used standard panel statistical analysis techniques to identify the impacts of irrigation on maize productivity via heat stress reduction and water stress reduction pathways.

Comparison of LST, ET, PET, ET/PET, GDD and EDD between irrigated and rainfed maize areas was performed within each county to minimize the effects of other spatially-varying factors, like background temperature and management practices, on surface temperature and evapotranspiration. These biophysical variables averaged over each county were then integrated over vegetative period (VP, from emergence date to silking date), grain filling period (GFP, from silking date to maturity date) and whole growing season (GS, from emergence date to maturity date) so we could evaluate whether and how irrigation had differentially influenced maize growth during early VP and late GFP.

We further examined how irrigation had changed the sensitivity of maize yield and its components to temperature variation. As done in our previous study (Zhu et al., 2019), we decomposed the total yield variation into three components: biomass growth rate (BGR), growing season length (GSL) and harvest index (HI) based on the following equation:

$$Yield = HI \cdot AGB = HI \cdot BGR \cdot GSL \quad (5)$$

Aboveground biomass (AGB) was retrieved through a regression model:

$$\mathrm{AGB} = 16.4 \cdot \mathrm{IWDRVI}^{0.8} \quad (6)$$

which was built in the previous study through regressing field measured maize AGB against MODIS derived integrated WDRVI (IWDRVI) (Zhu et al., 2019). Then HI could be estimated as Yield/AGB and BGR could be estimated as AGB/GSL. Such decomposition allowed us to examine how different crop growth physiological processes responded to external forcing: HI characterizes dry matter partitioning between source organ and sink organ and is mainly related with processes determining grain size and grain weight; BGR is related with physiological processes of daily carbon assimilation rate through photosynthesis and GSL is related with crop phenological development. The uncertainties related with AGB estimation was quantified through resampling as we did in previous studies (Zhu et al., 2019).

Temperature sensitivity of irrigated or rainfed yield ($S_T^{Yield}$) was estimated using a panel data model (Eq. (7)) with growing season mean LST and ET/PET as the explanatory variables:

$$log(Yield_{i,t}) = \gamma_1 t + \gamma_2 LST_{i,t} + \gamma_3 \frac{ET}{PET}_{i,t} + County_i + \varepsilon_{i,t} \quad (7)$$

$Yield_{i,t}$ is maize yield (t/ha) in county i and year t. It was a function of overall yield trends ($\gamma_1 t$) that had fairly steadily increased over the study period (Figure 1b), local crop temperature stress ($LST_{i,t}$), and local crop water stress ($\frac{ET}{PET}_{i,t}$). The $County_i$ terms provided an independent intercept for each county (fixed effect), and thus accounted for time-invariant county-level differences that contributed to variations in yield, like the soil quality. $\varepsilon_{i,t}$ is an idiosyncratic error term. $\gamma_2$ or $\frac{\partial \ln(Yield)}{\partial LST}$ defines the temperature sensitivity of yield. The temperature sensitivity of BGR ($S_T^{BGR}$), HI ($S_T^{HI}$) and GSL ($S_T^{GSL}$) could be estimated with Eq (7) in a similar way through using BGR, HI and GSL as the dependent variable. Here the dependent variable Yield (BGR, GSL and HI) was logged, so the estimated temperature sensitivity represented the percentage change of Yield (BGR, GSL and HI) with 1 ℃ temperature increase.

To quantify the relative contribution of water and high temperature stress alleviation to yield benefit, the yield difference between irrigated and non-irrigated maize (irrigation yield-rainfed yield, $\Delta Yield$) was regressed over the quadratic function of growing season EDD and ET/PET differences between irrigated and rainfed maize:

$$\Delta Yield_{i,t} = \gamma_1 \Delta \frac{ET}{PET}_{i,t} + \gamma_2 \Delta \frac{ET}{PET}_{i,t}^{2} + \gamma_3 \Delta EDD_{i,t} + \gamma_4 \Delta EDD_{i,t}^2 + County_i + \varepsilon_{i,t} \quad (8)$$

The yield improvement explained by heat and water stress alleviation was estimated as $\frac{\gamma_1 \sum \Delta \frac{ET}{PET}_{i,t} + \gamma_2 \sum \Delta \frac{ET}{PET}_{i,t}^{2} + \gamma_3 \sum \Delta EDD_{i,t} + \gamma_4 \sum \Delta EDD_{i,t}^2}{\sum \Delta Yield_{i,t}}$. The relative contribution of water and high temperature stress alleviation was estimated as $\frac{\gamma_1 \sum \Delta \frac{ET}{PET}_{i,t} + \gamma_2 \sum \Delta \frac{ET}{PET}_{i,t}^{2}}{\gamma_1 \sum \Delta \frac{ET}{PET}_{i,t} + \gamma_2 \sum \Delta \frac{ET}{PET}_{i,t}^{2} + \gamma_3 \sum \Delta EDD_{i,t} + \gamma_4 \sum \Delta EDD_{i,t}^2}$ and

$\frac{\gamma_3 \sum \Delta EDD_{i,t} + \gamma_4 \sum \Delta EDD_{i,t}^2}{\gamma_1 \sum \Delta \frac{ET}{PET}_{i,t} + \gamma_2 \sum \Delta \frac{ET}{PET}_{i,t}^2 + \gamma_3 \sum \Delta EDD_{i,t} + \gamma_4 \sum \Delta EDD_{i,t}^2}$, respectively. Given the potential collinearity between $\Delta\frac{ET}{PET}$ and ΔEDD, we also calculated the Variance inflation factor (VIF) to diagnose the severity of collinearity. The daytime LST difference ($\Delta LST$) was also tested to characterize heat stress alleviation with the following equation:

$$\Delta Yield_{i,t} = \gamma_1 \Delta \frac{ET}{PET}_{i,t} + \gamma_2 \Delta \frac{ET}{PET}_{i,t}^2 + \gamma_3 \Delta LST_{i,t} + \gamma_4 \Delta LST_{i,t}^2 + County_i + \varepsilon_{i,t} \qquad (9)$$

Then, the relative contribution of water and high temperature stress alleviation was estimated as $\frac{\gamma_1 \sum \Delta \frac{ET}{PET}_{i,t} + \gamma_2 \sum \Delta \frac{ET}{PET}_{i,t}^2}{\gamma_1 \sum \Delta \frac{ET}{PET}_{i,t} + \gamma_2 \sum \Delta \frac{ET}{PET}_{i,t}^2 + \gamma_3 \sum \Delta LST_{i,t} + \gamma_4 \sum \Delta LST_{i,t}^2}$ and $\frac{\gamma_3 \sum \Delta LST_{i,t} + \gamma_4 \sum \Delta LST_{i,t}^2}{\gamma_1 \sum \Delta \frac{ET}{PET}_{i,t} + \gamma_2 \sum \Delta \frac{ET}{PET}_{i,t}^2 + \gamma_3 \sum \Delta LST_{i,t} + \gamma_4 \sum \Delta LST_{i,t}^2}$, respectively.

## 3. Results

As expected, irrigation improved maize yield and the yield benefit showed a distinct spatial variation when we compared areas we identified as irrigated versus rainfed maize. The yield benefit of irrigation was much higher in the western area of the state (Figure 2a), because the drier environment in western area widened the yield gap between irrigated and rainfed cropland in an average year. The satellite derived vegetation index WDRVI reflected these differences, with higher values in areas we identified as irrigated maize, especially around maize silking (Figure 2b). Importantly, this suggested that, in conjunction with ground-based information calibrated crop phenology, irrigated and rainfed cropland were distinguishable with time series satellite data where rainfall does not meet crop water demand.

When county-level LST data were averaged over 2003-2016, the daytime LST in irrigated maize was 1.5℃ cooler than rainfed maize, while nighttime LST showed a very slight difference (0.2 ℃) (Figure 3a,b). When the LST differences were

integrated over different growing periods (Figure 3e-h), we found that the daytime cooling effect was greatest in the GFP (Figure 3g), probably due to the higher LAI (or ground cover) and transpiration during that stage of growth. This was also consistent with previous field studies showing that irrigation was mainly applied during the middle to late reproductive period, which corresponded to the greatest water demand period (Chen et al., 2018). The spatial pattern of the LST difference showed stronger cooling effect in the western area (Figure 3c-h), which was similar to the spatial pattern of yield benefit identified in Figure 2a. In contrast, surface air temperature shows much smaller daytime cooling effect (Figure 3i,j). The mean air temperature difference between irrigated and rainfed maize in daytime and nighttime were −0.2℃ and −0.3℃, respectively, and the spatial pattern of air temperature difference over VP and GFP was also relatively small between counties and crop growth periods (Figure 3k-p).

Temperature is an important driver of crop phenology and has been used as the primary environmental variables in crop phenology models (Wang et al., 1998). Given the identified irrigation cooling, we further looked into how irrigation altered maize phenological stages. We found irrigated maize showed an earlier emergence and silking but delayed maturity (Figure 4a). Consequently, GFP was extended by 7.5 days on average, which contributed to most of GS extension (8.1 days) (Figure 4b). Site measurements of phenological stage information confirmed that irrigated maize had a longer GS, especially during GFP (Figure 4c). The reason why such extension mainly occurred in GFP might be that (1) LST cooling was more prominent during GFP and (2) phenological development during GFP was more sensitive to temperature variation than development during VP (Egli et al., 2004). The higher temperature sensitivity of phenological development during GFP (4.9 day/℃) was confirmed when we regress GFP difference between irrigated and rainfed maize over LST difference between irrigated and rainfed maize (Figure 4d-f). The spatial pattern suggested GS and GFP extension was more significant in the western area (Figure 4g-h), likely due to the corresponding stronger cooling effect.

We integrated LST or air temperature as described above (Materials and Methods) to estimate heat exposure (GDD and EDD) over maize growing season. We found both LST and air temperature estimated GDD were greater in irrigated maize than GDD in

rainfed maize across most counties, especially during GFP (Figure 5a,c), which was very likely due to the GFP extension. As GDD characterizes the beneficial thermal time accumulation, the greater GDD in irrigated maize might contribute to the higher yield. In terms of EDD, LST estimated EDD suggested that irrigation suppressed high temperature stress especially for GFP (Figure 5b), while air temperature estimated EDD failed to characterize the irrigation induced lower high temperature stress (Figure 5d).

SSEBop ET and MODIS PET were used to explore how irrigation influenced water demand and water supply across maize. We found irrigation led to 27% higher ET and 2% lower PET (Figure 6a-b). Higher ET was anticipated in irrigated maize, and lower PET might be due to irrigation cooling effect, which resulted in lower VPD and thus lower evaporative demand. We used the ratio of ET to PET as the metric for water stress in this study, where low values indicated that plants were not transpiring at their full potential in the ambient conditions. This ratio was higher for irrigated maize, especially during the GFP (Figure 6c), and the spatial distribution suggested that the difference was greater in western counties than eastern counties (Figure 6d-e), which was similar to the distribution of the local cooling effect identified in Figure 3c.

We divided the temperature sensitivity of yield into three components (sensitivity of BGR, GSL and HI) to investigate how irrigation changed the response of maize physiological processes to temperature. As shown in Figure 7, we found that temperature sensitivity of yield was significantly weakened from $-6.9\%/^{\circ}C$ to $-1\%/^{\circ}C$ in irrigated vs. rainfed areas, and this yield sensitivity change was mainly driven by a change in the sensitivity of the HI, which was weakened from $-4.2\%/^{\circ}C$ to $1\%/^{\circ}C$. In both rainfed and irrigated maize, temperature sensitivity of GSL was quite close (approximately $-2\%/^{\circ}C$), while BGR was only slightly influenced by temperature (Figure 7).

We found that irrigation application not only lowered water and high temperature stress, but also made yield less sensitive to water and high temperature stress (Figure 8a-c), consistent with previous studies (Troy et al., 2015; Tack et al., 2017). We regressed yield differences over climatic variables differences using the linear model

(Eq. (8)), and estimated that 61% of yield improvement between irrigated and rainfed maize could be explained by the irrigation induced heat and water stress alleviation. We further calculated that 79% of yield improvement was due to water stress alleviation and 21% due to heat stress alleviation. Because the distribution of $\Delta EDD$ was truncated for points with $\Delta EDD > 0$ (Figure 8e), we explored an alternative model with quadratic functions of $\Delta LST$ and $\Delta ET/PET$ (Eq. (9)). In this specification, 72% of yield improvement can be explained by water and high temperature stress alleviation, with 65% and 35% of yield improvement due to water and high temperature stress alleviation, respectively. The VIF we used to diagnose the collinearity between $\Delta LST$ and $\Delta ET/PET$ was 2.2. Normally, VIFs over 10 indicate collinear variables (with 5 being a more strict standard), therefore, our VIF test suggested the collinearity was not severe, probably because we used differences of LST and $ET/PET$ between irrigated and rainfed maize rather than directly using LST and $ET/PET$ as the explanatory variables.

Because we found a strong effect on yields via the heat stress (and not simply water stress), we compared our results with four process-based crop models that simulated crop growth under both rainfed and irrigated conditions. These simulations qualitatively reproduced the irrigation-induced higher maize yield, biomass, and ET (Figure 9), but to different degrees. The highest modeled improvement was identified in CLM-crop, with an increase of 57%, 43% and 32% in yield, biomass and ET, respectively. However, all models except CLM-crop failed to reproduce the growing stage extension under irrigation (Figure 9), probably because only CLM-crop implemented canopy energy balance module to simulate canopy temperature. CLM-crop was thus the only model able to capture the irrigation-induced evaporative cooling effect (the heat-stress reduction). That the best agreement between observed and modeled results occurred with the only model that plausibly accounted for heat-stress alleviation due to irrigation was further evidence that this was the phenomenon we captured in our satellite observational study.

## 4. Discussion and conclusion

By integrating satellite products and ground-based information about cropping and irrigation, we showed that irrigated maize yields were higher than rainfed maize

yields because added irrigation water reduced heat stress in addition to water stress. Our study underlines the relative importance of heat stress alleviation in yield improvement and the necessity of incorporating crop canopy temperature models to better characterize heat stress impacts on crop yields (Teixeira et al., 2013; Kar and Kumar, 2007). Our analysis disentangling the relative importance of heat and water stress alleviation in yield benefit helps farmers plan future investments, especially in terms of selecting cultivars with heat or drought stress tolerance. In addition, disentangling the two effects allows crop models better predict crop phenology, considering irrigation induced cooling effect alters maize growing phases.

Although ours is not the first study to suggest replace air temperature with MODIS LST for maize yield prediction, especially under extreme warm and dry conditions, our results underscore important implications of doing so. Given the important role of heat stress in determining crop yield, thermal band derived LST information at finer spatial and temporal resolution should be a critical input for satellite data driven yield prediction models (Wang et al., 2015; Huryna et al., 2019; Li et al., 2019; Meerdink et al., 2019). In addition, given the differential responses of crop growth to heat and water stresses in different stages, fusing satellite derived crop stage information with the heat and water stressors might improve crop yield prediction.

This study also has useful implications for process-based crop model development. In our model evaluation procedure, only one model has implemented canopy energy balance scheme, but it is the one that captures the observed maize growth stage extension. Our results suggest that the heat stress alleviation due to irrigation identified here is largely overlooked in current crop models. As such, when those crop models are calibrated to match observed yields, processes associated with water stress alleviation are probably overestimated, resulting in uncertainties for predicting future irrigation water demand and crop yield. These uncertainties might mislead future adaptation decisions due to incomplete or biased estimates of the relative contributions of heat and water stress. Relatedly, recent studies identified a wide range for the simulated canopy temperature in current crop models (Webber et al., 2017). Therefore, assimilating satellite derived LST might be a potential solution to improving heat stress representation (Meng et al., 2009; Xu et al., 2011), especially given that the recent ECOsystem Spaceborne Thermal Radiometer Experiment on

Space Station (ECOSTRESS) mission makes hourly plant temperature measurement available (Meerdink et al., 2019).

Several limitations and caveats apply to our study. First, the daily MODIS daytime LST we used to explain crop maximum daily temperature had missing value due to quality control and was derived from a mix of crop covers and other land surface temperature information, which might bias the identified irrigation cooling effect. Specifically, using MODIS daytime LST as a proxy for true (measured) maximum crop surface temperature in an empirical statistical model might underestimate the benefit of cooling effect (measurement error in a predictor variable producing attenuation bias). These uncertainties in LST dataset might be resolved with the recently launched ECOSTRESS mission, as its hourly revisiting frequency enables better estimation of maximum daily temperature. The second issue is that water stress and heat stress were not perfectly separable. As what we have shown, the cooling effect of irrigation lowers evaporative demand (PET) and thus indirectly contributes to lower water stress (higher ET/PET). Our disentangling method do not account for the water stress and heat stress interaction effects. If these interaction effects were considered, the relative contribution of heat stress alleviation to yield improvement is likely to be higher. Such water and heat stress interactions can become stronger during extreme years, like 2012, when most of the Midwest experienced severe drought. Under such circumstances, irrigation-induced cooling effect will be more beneficial. The third issue is that our study is conducted for maize in only one state, Nebraska. Although Nebraska is the largest irrigation maize producer in the US, results might differ for other crop types and other landscapes, due to different crop canopy structures and management practices (Chen et al., 2018), and spatial variations in water and heat stresses mitigation effects (Figure 3 and Figure 7).

Overall, our study suggests that heat stress alleviation, in addition to water stress alleviation, plays an important role in improving irrigated maize yield. Since current models generally cannot accurately simulate the canopy temperature, the irrigation induced yield benefit might have been overly attributed to water stress alleviation. This might bias the future yield prediction under irrigation, since high temperature stress might be more dominant than drought for crop yield formation under future warmer climate (Zhu et al, 2019; Jin et al., 2017). Therefore, better constrained crop

models through integrating satellite observed land surface temperature and crop stage information will be necessary to improve yield prediction and help policymakers and farmers make better decisions on where and when to implement irrigation.

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

## Acknowledgments

We thank the NSF/USDA NIFA INFEWS T1 #1639318 for funding support.

Figures

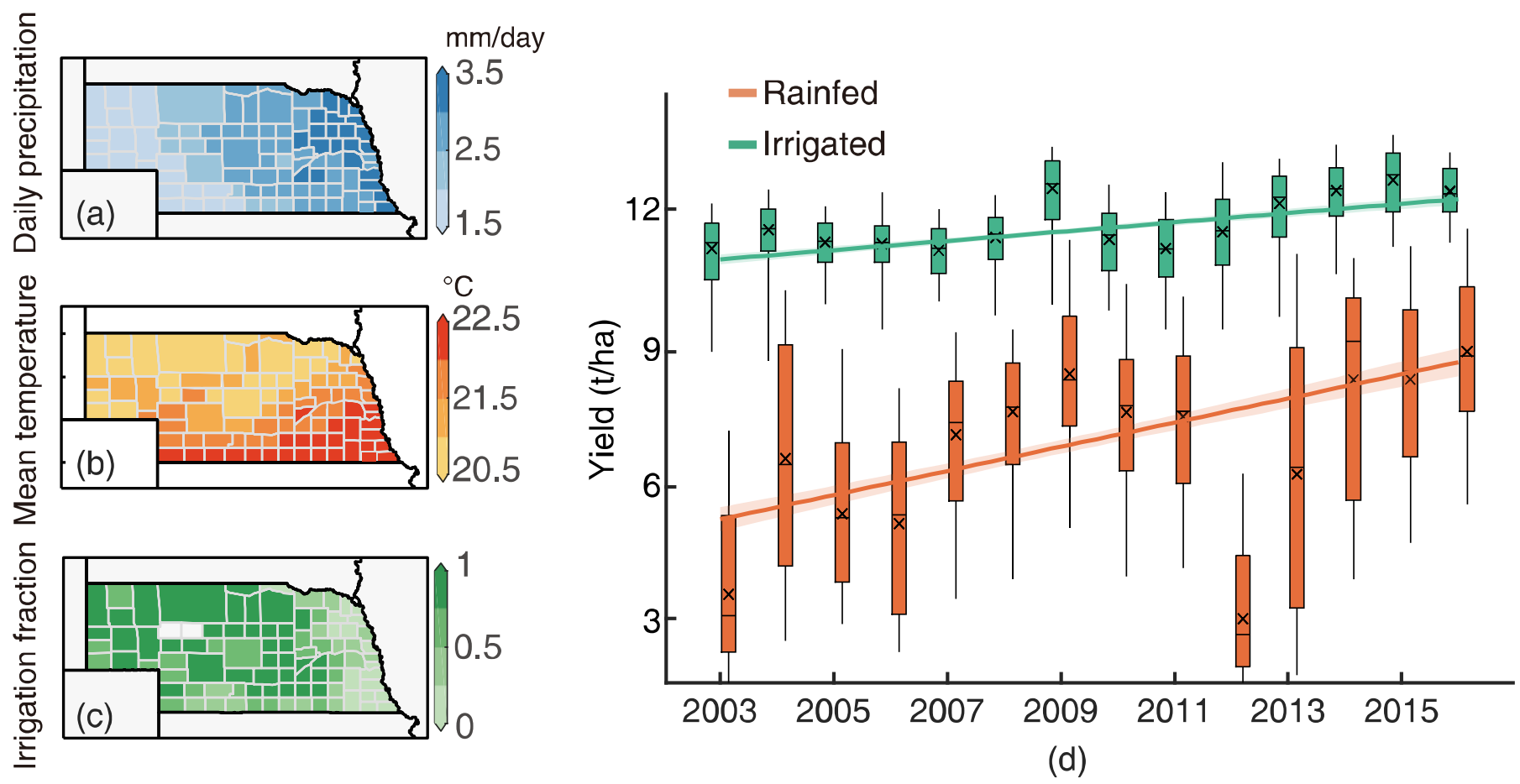

**Figure 1:** The spatial pattern of county level multi-year (2003-2016) mean daily precipitation (a) and air temperature (b) during maize growing season. County level multi-year (2003-2016) mean maize irrigation fraction across Nebraska (c). The maize irrigation fraction is based on USDA NASS report. Boxplot of county level irrigated and rainfed maize yield in Nebraska over the study period (d). The lines in (d) show the linear fitted yield trend with 95% confidence interval. Boxplots indicate the median (horizontal line), mean (cross), inter-quartile range (box), and 5–95th percentile (whiskers) of rainfed or irrigated yield across all counties.

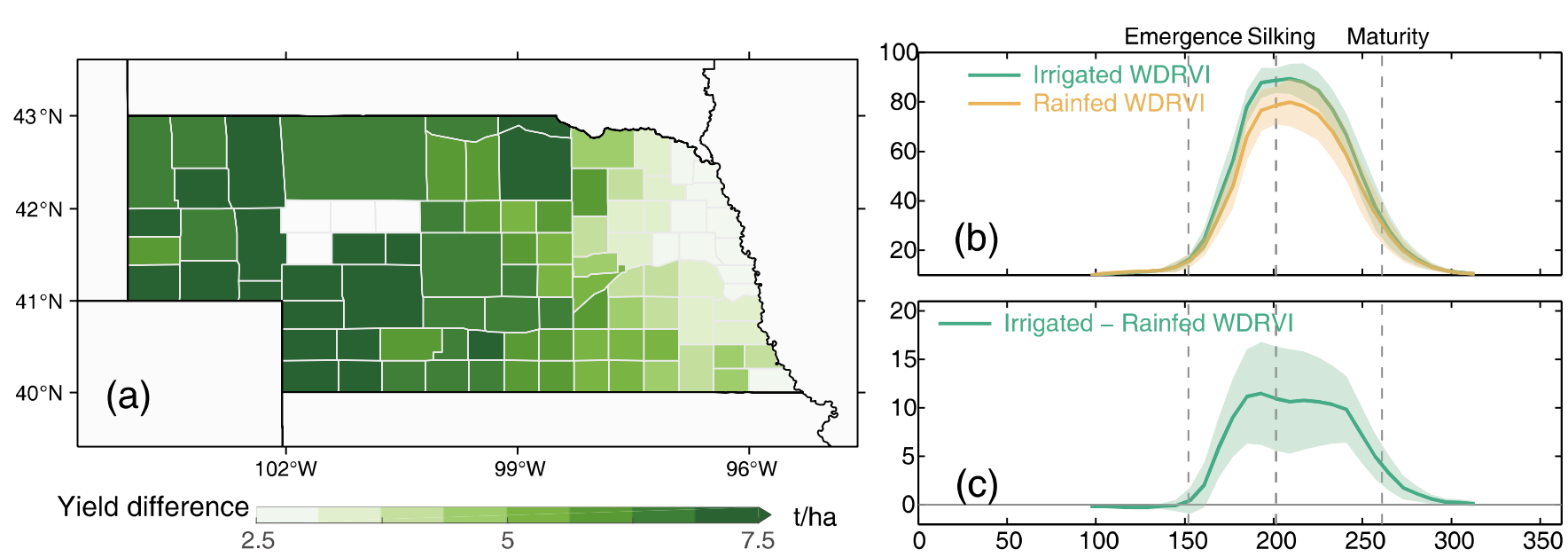

**Figure 2:** The difference between irrigated and rainfed maize yield (a) and satellite observed vegetation index (b and c). The shaded area in (b) and (c) shows one standard deviation of WDRVI (b) and WDRVI difference (c).

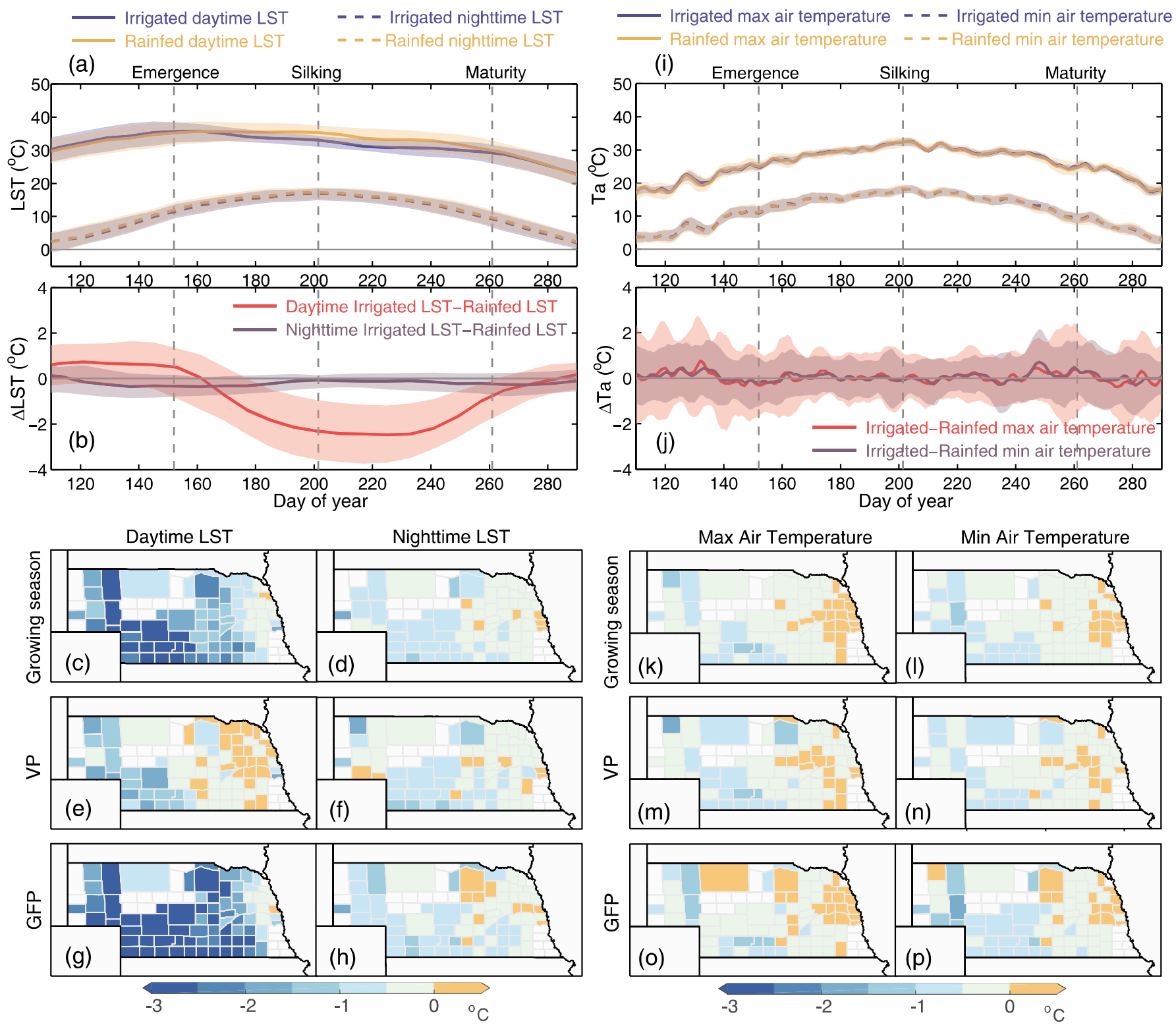

**Figure 3:** Spatial-temporal patterns of daytime and nighttime MODIS LST differences (left panel, a-h) and surface air temperature differences (right panel, i-p) between irrigated and rainfed maize in different growth stages: vegetative period and grain filling period. The shaded areas in (a), (b) and (i), (j) show one standard deviation of corresponding variables.

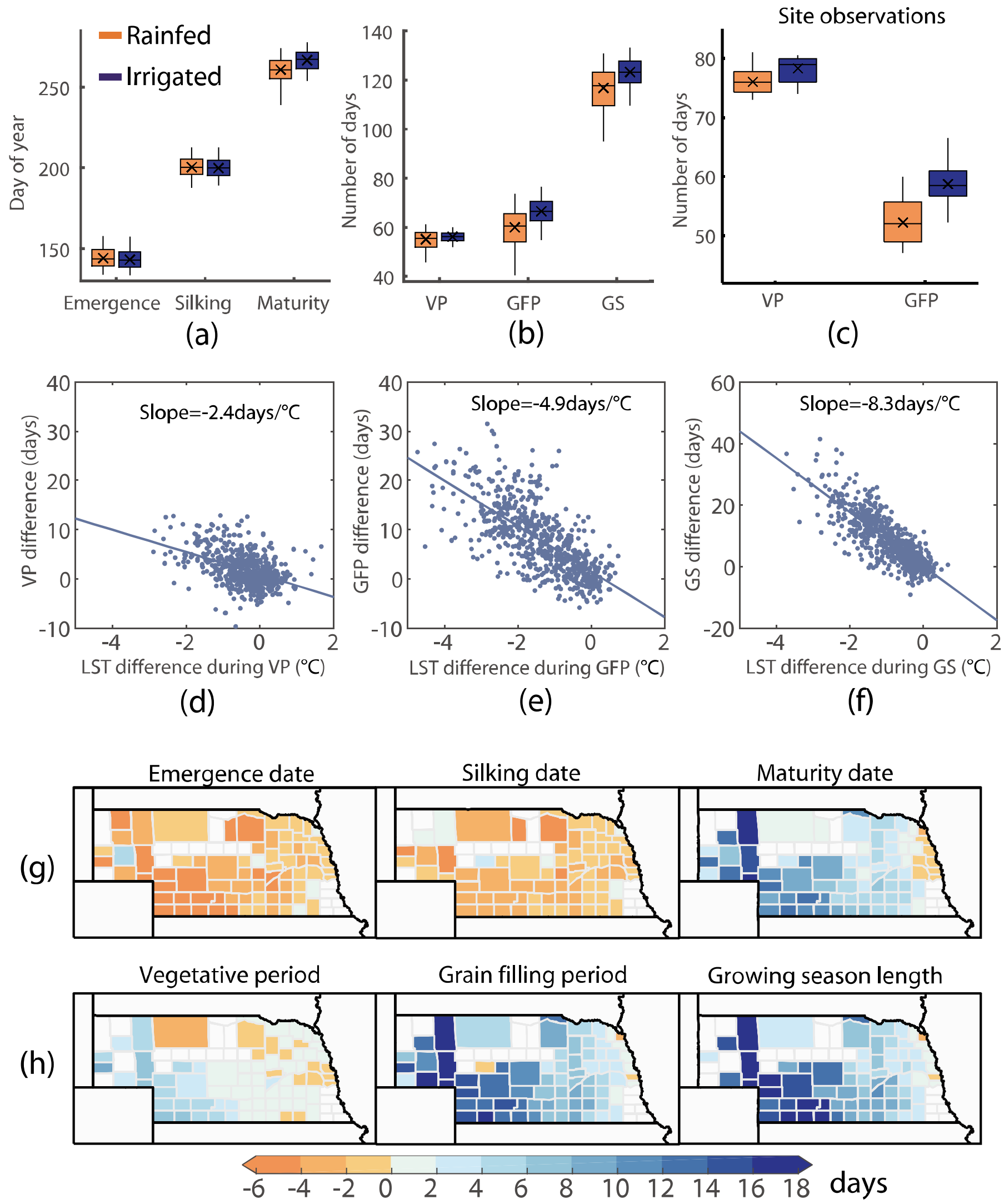

**Figure 4:** Boxplot of maize phenological date (a) and duration (b-c) for irrigated and rainfed maize areas. Sensitivity of phenological duration difference between irrigated and rainfed maize to LST difference between irrigated and rainfed maize (d-f). The slope in (d-f) was estimated with linear model. The spatial pattern of phenological date and duration differences between irrigated and rainfed maize areas (g-h).

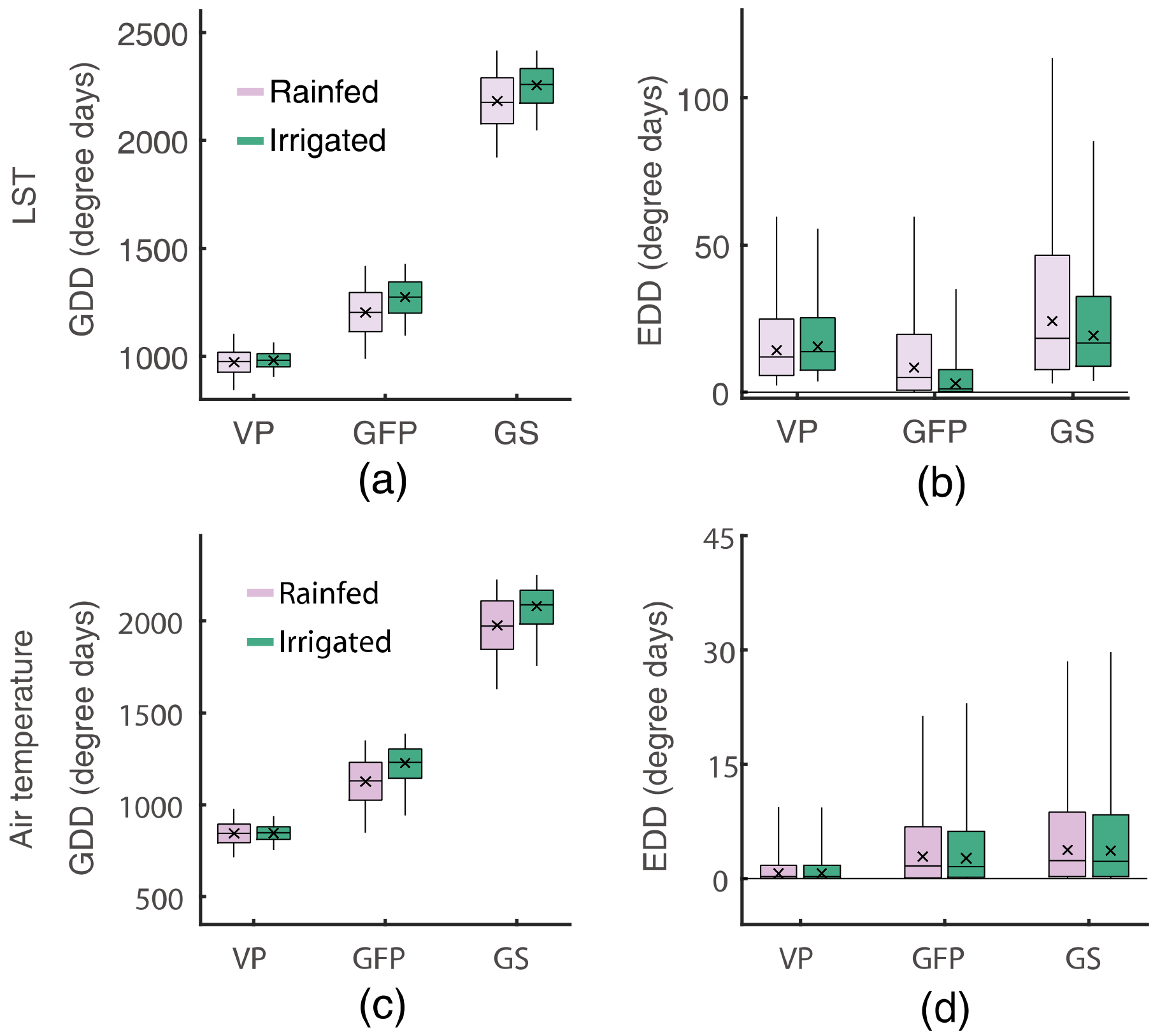

**Figure 5:** Boxplot of GDD and EDD estimated with MODIS LST (a-b) and surface air temperature (c-d) for irrigated and rainfed maize areas. Boxplots indicate the mean (cross), median (horizontal line), 25--75th percentile (box), and 5--95th percentile (whiskers) of corresponding variables in all year and county combinations.

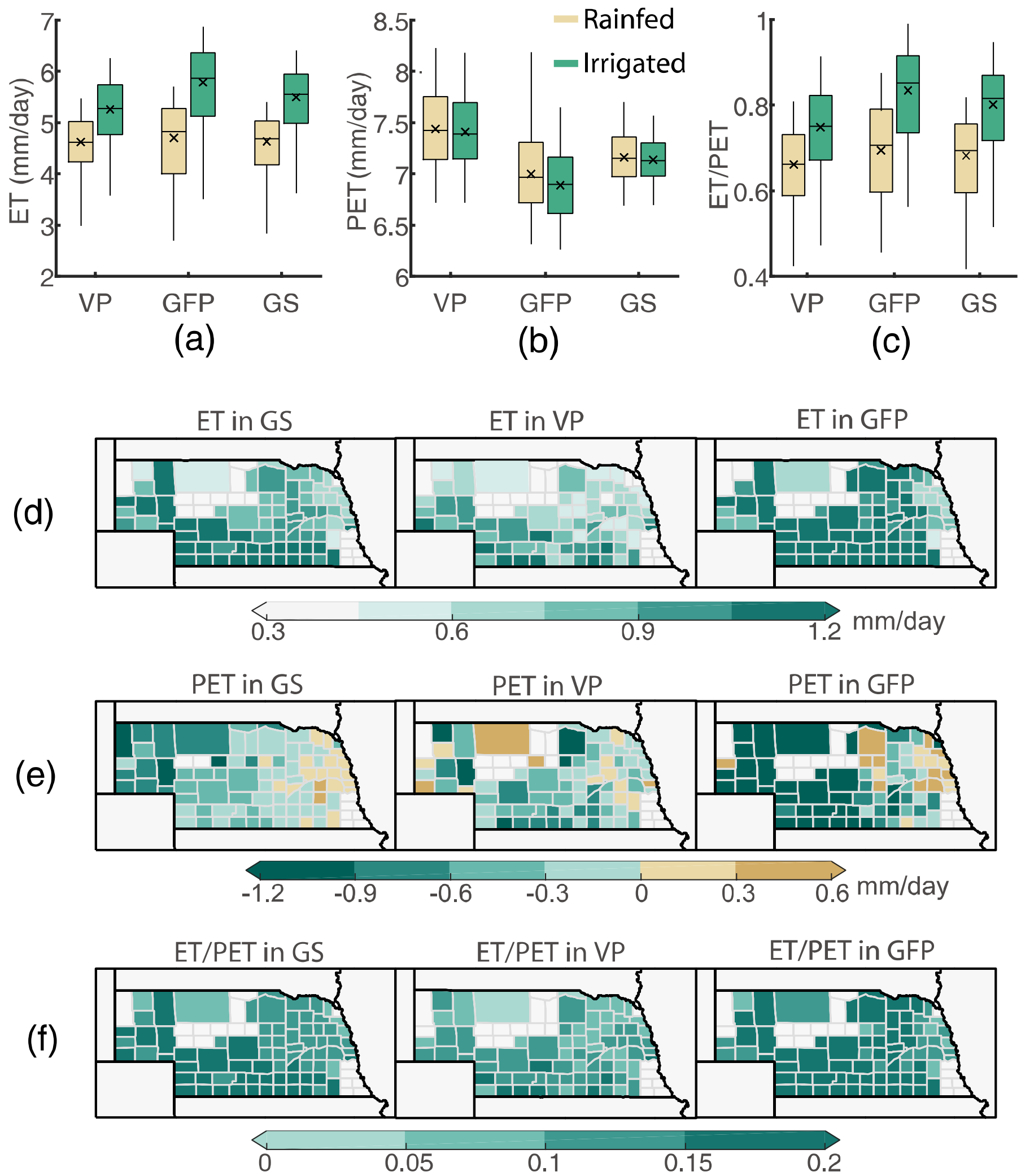

**Figure 6:** Boxplot of SSEBop ET, MODIS PET and ET/PET for irrigated and rainfed maize areas (a-c). Spatial pattern of SSEBop ET, MODIS PET and ET/PET differences between irrigated and rainfed maize areas (d-f).

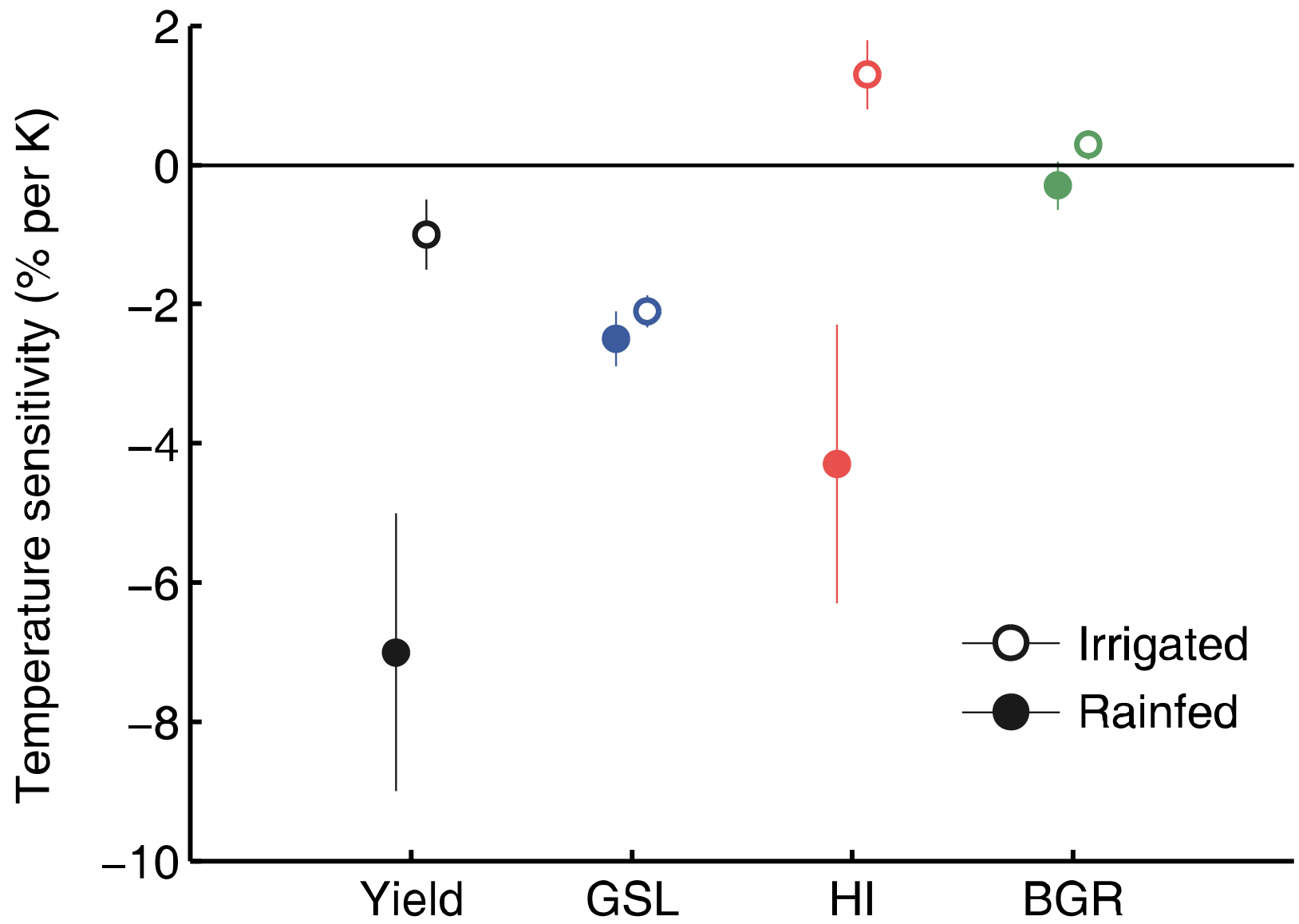

**Figure 7:** Temperature sensitivity of yield and yield components (GSL, HI and BGR) for irrigated and rainfed maize areas.

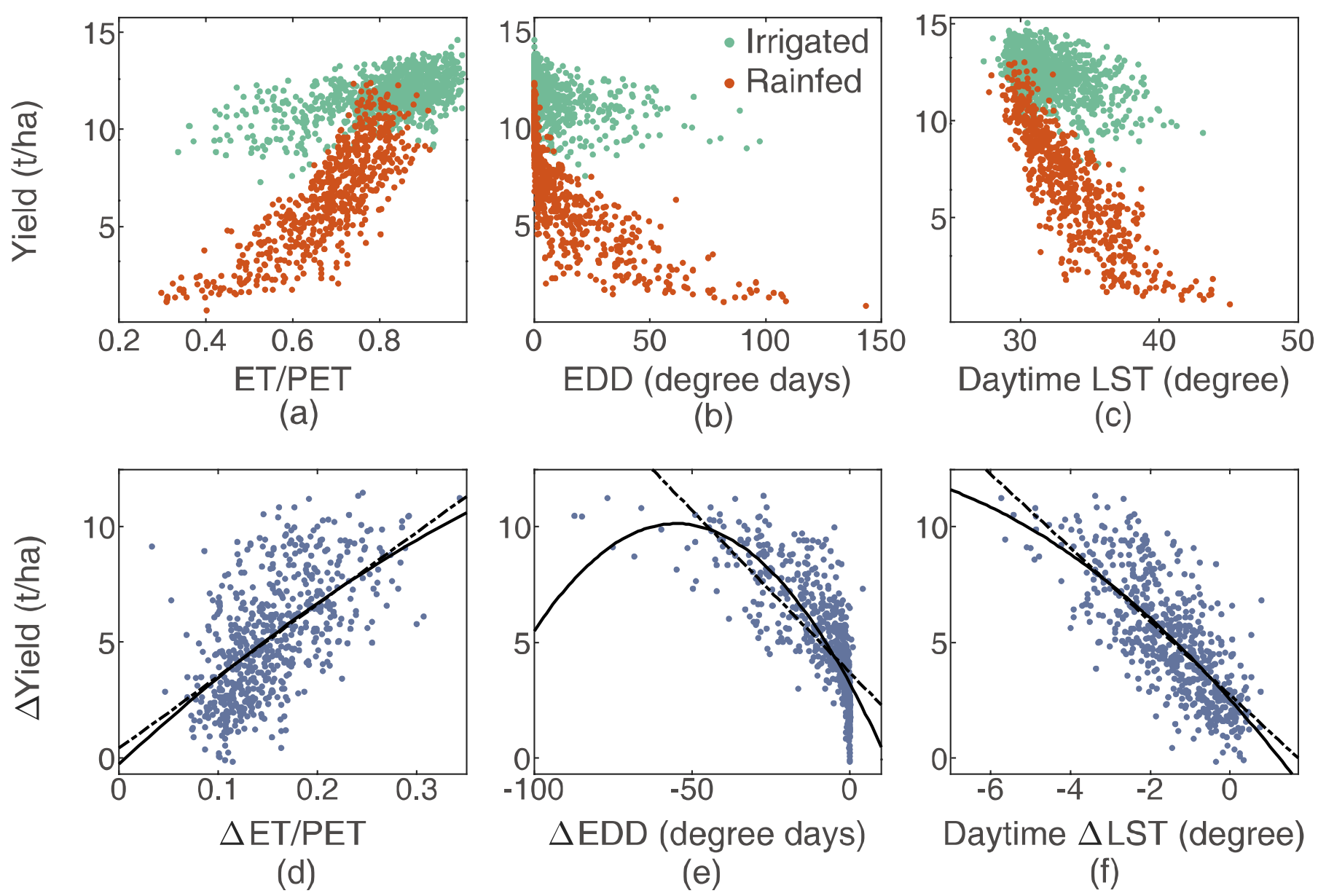

**Figure 8:** Response of maize yield to ET/PET (a), EDD (b) and daytime LST (c) in both irrigated and rainfed maize. Response of yield differences to ET/PET (d), EDD (e) and daytime LST (f) differences between irrigated and rainfed maize. The linear (dash black line) and quadratic (solid black line) response curves of $\Delta Yield$ to $\Delta ET/PET$, $\Delta EDD$ and $\Delta LST$ are shown in d-f.

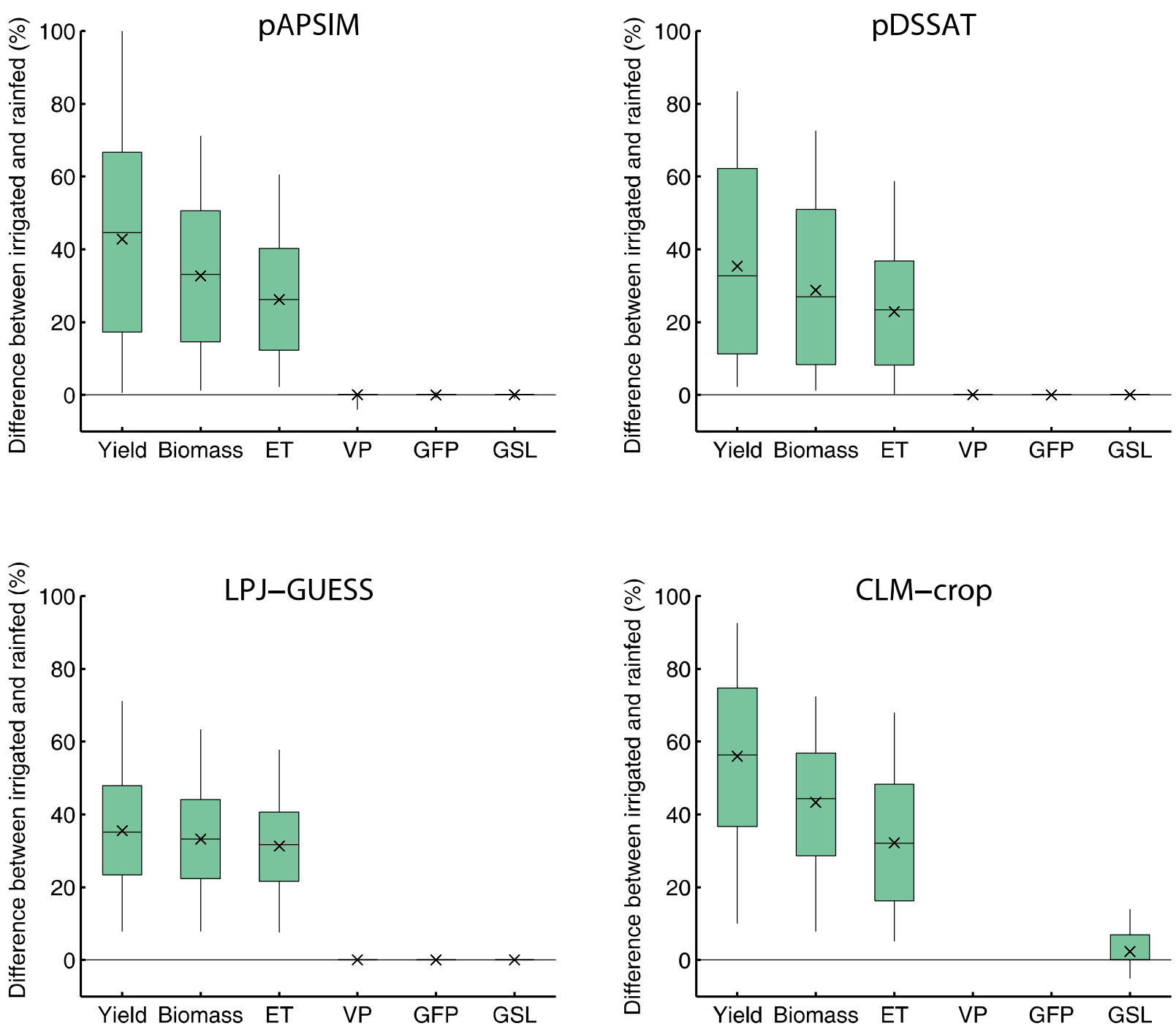

**Figure 9:** Boxplot of crop model simulated yield, biomass, ET and phenological duration (VP, GFP and GSL) differences between irrigated and rainfed maize areas. For phenological duration, CLM-crop only reports GSL.