# Peer review of "Untangling irrigation effects on maize water and heat stress"

_Hydrology and Earth System Sciences, 2020_

## Author Comment (AC1)

Review1:

This is an interesting article describing effects of high temperature and drought on maize yield and yield components in Nebraska. The authors used remote sensing to detect high temperature stress and drought stress and also tested whether four different crop models can reproduce the effects detected by remote sensing. The article is well written, good to understand and figures are of high quality. However, I cannot recommend to publish the present version of the article in HESS. My major criticisms are:

We thank the reviewer for constructive evaluations and suggestions. We have accordingly revised our manuscript following the reviewer's suggestions, as detailed in the point-by-point response below.

1) The major source to describe high temperature and drought stress in maize are land surface temperature and ET detected by remote sensing. I think that the temperature based indicators LST and EDD are highly determined by the ratio ET/PET which was used to describe drought impacts. Which factor different from drought can explain canopy temperature differences between well watered and rainfed maize fields? Or in other words: can differences in LST and EDD at the same location happen independently of drought stress? I don't think so. If so, for example because of different LAI, then this is likely an affect of drought in previous growth stages.

It is well understood that transpiration cooling is directly controlled by the stomata conductance and vapor pressure deficit, which are again controlled by drought. This is also the reason why canopy temperature differences are often used as indicator for drought stress or even for irrigation scheduling. Consequently I think that EDD differences or LST differences between irrigated and rainfed maize in the same region are just another manifestation of differences in drought stress between irrigated and rainfed fields. From that perspective I cannot understand why the collinearity tests performed for the variables included in equation 7-9 did not show critical values.

Thank you for this comment – this is the very heart of the research question at hand: how much of temperature stress is really moisture stress (and vice versa)? While we understand your intuition that extreme heat is hurting plants exclusively through moisture (drought) stress, there is in fact significant variation in the sample between EDD and ET/PET. We show this in the collinearity tests and in the figure below.

[Figure]

The left panel is the original growing season mean ET/PET and LST, the right panel is the difference of LST ($\Delta$LST) and ET/PET ($\Delta$ET/PET) between rainfed and irrigated maize.

Although the difference of ET/PET and LST was still correlated, we also calculated the Variance inflation factor (VIF) to diagnose the severity of collinearity and the impact it might be having on the ability to statistically estimate the parameters of interest. The statistical rule-of-thumb is that VIFs over 10 indicate collinear variables that may be causing coefficient instability (with 5 being a more strict standard). Here the VIF in our statistical model is 2.2, suggesting the collinearity is not severe, and there remains enough independent variation to trust our estimates. The results may be surprising, but these two parameters do in fact exhibit some independent variation. This is why the variance inflation factors are quite low (and thus not problematic for the statistical estimation).

In sum, we understand the reviewer's intuition and potential surprise at the amount of independence here; this is the very reason we think this is an exciting and publishable result.

2) The authors showed that there are considerable differences in the growing season length of irrigated and rainfed maize and suggest that the differences are mainly an effect of cooler canopy temperature under well watered conditions (lines 322-337). Another potential reason could be the so called drought escape effect. It is known that many crops speed up their phenological development under drought to make sure that grains reach physiological maturity before the stress becomes so strong that the crop has to die. Again, in that case it would be a drought effect and not an effect of higher temperatures. I agree that it is not so easy to find out which effect really matters. I suggest to test the GDD computed in equation 3 for years with similar canopy temperature but different drought stress (ET/PET ratio). For example, a year that is warm and wet should result in similar canopy temperatures compared to a year that is a bit cooler but dry. Important is that the test has to be made for the same location (county) to avoid that cultivar differences between warmer and cooler regions disturb the relationship. If for years with similar canopy temperature but different ET/PET ratio the GDD is similar, then the shorting of the growing period is independently of drought and the drought escape mechanism can be excluded. If GDD is, for similar canopy temperatures, positively correlated with the ET/PET ratio, then this would point to the drought escape mechanism.

Thanks for your suggestion. We do a test based on the reviewer's suggestion. For each county, we calculated the difference of GDD between rainfed and irrigated and then plotted it against the difference of ET/PET and LST, respectively. This method ensures

that the comparison is conducted for the same county to minimize the effects of cultivar differences, as the reviewer suggested. As the following figure shows, we can see there is a clear decline in ΔGDD with higher LST but no significant decline in ΔGDD with lower ET/PET.

[Figure]

**Specific comments:**

Line 175 (equation 3): Why was it decided to set the high temperature threshold to 30 dC? In the literature heat stress thresholds for maize are typically higher, about 34 dC (Sanchez et al., 2014).

Indeed there are various ways to define high temperature stress. In addition to the suggestion by the reviewer to use 34°C, as done by Sanchez et al., (2014), there are also some studies using 30 °C (Lobell et al 2011; Lobell et al., 2013; Zhu et al., 2019) or 29°C (Butler et al., 2018). So we believe the threshold we used to define high temperature stress is defensible. We also include a clarification for this point with the suggested reference added in line 180: "Following previous studies (Lobell et al., 2011; Zhu et al., 2019), 30°C is set as the high temperature threshold, although higher value might be also applicable as the temperature threshold (Sanchez et al., 2014)."

Butler E E, Mueller N D, Huybers P. Peculiarly pleasant weather for US maize[J]. Proceedings of the National Academy of Sciences, 2018, 115(47): 11935-11940.

Lobell D B, Bänziger M, Magorokosho C, et al. Nonlinear heat effects on African maize as evidenced by historical yield trials[J]. Nature climate change, 2011, 1(1): 42-45.

Lobell D B, Hammer G L, McLean G, et al. The critical role of extreme heat for maize production in the United States[J]. Nature climate change, 2013, 3(5): 497-501.

Zhu P, Zhuang Q, Archontoulis S V, et al. Dissecting the nonlinear response of maize yield to high temperature stress with model‑data integration[J]. Global change biology, 2019, 25(7): 2470-2484.

Line 262 (equation 7): How was LST and ET/PET computed? As mean for the whole growing period? In the variable explanation (line 265) you call LST "local crop temperature stress" but shouldn't you then better use EDD here?

Thank you for pointing out the need for clarity here. We have edited line 202 to more clearly convey the variable construction: "Then ET, PET and ET/PET were averaged over time to get mean ET, PET and ET/PET during VP, GFP and GS with satellite derived phenology to characterize water status during maize growth." Here our main purpose is to quantify the temperature sensitivity of irrigated or rainfed yield, so we used LST rather than EDD as the explanatory variable.

Lines 280-292: Any reason why delta EDD and delta ET/PET are NOT highly correlated?

We refer the reviewer to our answer above. Again, this may be somewhat surprising, but it is indeed the case that they exhibit independent variation that enables our analysis.

Lines 363-365: "As shown in Figure 7, we found that temperature sensitivity of yield was significantly weakened from − 6.9%/â„ƒ to −1%/â„ƒ in irrigated vs. rainfed areas ..."

=> shouldn't this be vice versa (lower sensitivity in irrigated maize)?

Thank you for pointing out this. Yes, it should be "we found that temperature sensitivity of yield was significantly weakened from −6.9%/℃ to −1%/℃ in rainfed vs. irrigated areas"

Lines 438-442: The assimilation of satellite derived LST might in fact reduce crop model uncertainty but this helps only when LST data are available. Crop models are also often used for climate change impact analysis but for simulation of potential futures LST is not available. Another disadvantage could be that LST is sensor and satellite specific, for example due to the different overpass times. Therefore another recommendation could be to improve crop models so that they can reproduce the effects that were found in the present study and use remotely sensed LST for validation.

Thank you for the insightful comments. We incorporated them in line 445: "Therefore, assimilating satellite derived LST might be a potential solution to improving crop models heat stress representation so that they can better reproduce the observed heat stress effects (Meng et al., 2009; Xu et al., 2011). These remotely sensed LST can also be used to validate model simulated LST, especially given that the recent ECOsystem Spaceborne Thermal Radiometer Experiment on Space Station (ECOSTRESS) mission makes hourly plant temperature measurement available (Meerdink et al., 2019). However, it is worth noting that the availability of satellite LST presents a constraint when thinking about future climate change impact studies

In addition, some caution is required for validating model-simulated LST, since LST is sensor and satellite specific.".

Figure 8: It seems that there is also considerable drought stress in irrigated maize because the ET/PET ratio is often much lower than 1. Any explanation why yield under irrigated conditions is often much higher for similar ET/PET ratios? Because irrigated maize is more often grown in cooler regions?

This illustrates the main point of the paper. As the previous figure shows, there is some degree of decoupling between ET/PET and LST. That means irrigation is relieving the pure heat stress (not only heat-through-moisture stress) component, so for the same ET/PET ratio, the yield is higher in irrigated areas.

References:

Sanchez, B., Rasmussen, A. and Porter, J.R. (2014). Temperatures and the growth and development of maize and rice: a review. Global Change Biology 20, 408–417.

---

## Author Comment (AC3)

Review2:

Irrigation benefits crop yield mainly through water stress and high temperature stress mitigation. Although water stress alleviation via irrigation has been addressed intensively, a further understanding of high temperature stress alleviation is still required. This paper attempts to separate the irrigation effects on maize heat stress from that on water stress using satellite vegetation, temperature, and evaporation products. The paper is well written and structured overall. The conclusions are drawn based on solid analyses and interpretations of the results. The pathway provided to improve the state-of-the-art crop models is of great interest to the community.

The only major concern for me is the collinearity between LST and ET considering LST is directly impacted by ET through surface energy balance. Thus, it should be careful to disentangle the heat stress and water stress. More illustrations would be required.

We thank the reviewer for the constructive review and suggestions. Please see the point-by-point response below.

The following figure shows ET/PET v.s. LST in each county.

[Figure]

The left panel is the original growing season mean ET/PET and LST, the right panel is the difference of LST (ΔLST) and ET/PET (ΔET/PET) between rainfed and irrigated maize.

Although the difference of ET/PET and LST was still correlated, we also calculated the Variance inflation factor (VIF) to diagnose the severity of collinearity and the impact it might be having on the ability to statistically estimate the parameters of interest. The statistical rule-of-thumb is that VIFs over 10 indicate collinear variables that may be causing coefficient instability (with 5 being a more strict standard). Here the VIF in our statistical model is 2.2, suggesting the collinearity is not severe, and there remains enough independent variation to trust our estimates. The results may be surprising, but these two parameters do in fact exhibit some independent variation. This is why the variance inflation factors are quite low (and thus not problematic for the statistical estimation).

In sum, we understand the reviewer's intuition and potential surprise at the amount of independence here; this is the very reason we think this is an exciting and publishable result.

Some minor comments are as follows.

Lines 190-194, I could not find the citations 'Senay et al., 2013' and 'Velpuri et al., 2013' in the reference.

Thanks for catching the missing citations. We have added the two omitted references.

Lines 195-196, Is the PET also available in the SSEBop ET product? It would be better to use consistent ET and PET to calculate the water stress index.

SSEBop only has estimation for ET. But based on our validation using Flux tower estimations in Supplementary Figure 3-5, MODIS PET has a good performance. So we just blended the two products for our analysis.

Lines 256-257, I would suggest to simply describe the uncertainties of AGE rather than just include the reference for better readability.

We added some details on how we quantify the uncertainties related with AGB: "The uncertainties in AGB estimation results from the parameters in the regression model (Eq. (6)) converting IWDRVI to AGB. Here we quantified the uncertainties rooted in the estimated parameters through running the panel model 1000 times with the samples generated from each parameter's 95% confidence interval (Zhu et al., 2019)."

Lines 286-287, what is the added value by using daytime LST difference considering the relative contribution of water and high temperature stress alleviation to yield benefit has been analyzed using Eq. 8.

We have clarified the added value of using daytime LST as the explanatory variables in line 385: "Because the distribution of ΔEDD was truncated for points with ΔEDD> 0 (Figure 8e), we explored an alternative model with quadratic functions of ΔLST and ΔET/PET (Eq. (9))."

Lines 317-318 and lines 346-347, could some explanations be found for the different performances between LST and air temperature?

We added an explanation for the identified different performances between LST and air temperature in line 324: "The difference between spatial-temporal patterns identified using LST and air temperature was mainly because LST reflects canopy energy partition between latent heat flux and sensible heat flux. Additional moisture provided by irrigation makes more heat transported as latent heat flux, resulting in a cooling effect."

---

## Author Comment (AC5)

Review3:

This paper proposes, based on a series of LST measurements derived from Modis, to analyse the effects of thermal and water stress by considering a big data set over a large territory and a long time series 2003-2016. The originality of this work, beyond the considerable corpus of data, is to analyse the contributions of both types of stress.

 Thank you for this detailed and constructive review. We have made the suggested changes, as detailed in the point-by-point response below.

The approach nevertheless presents an important methodological flaw in the separation of thermal and hydric effects. Indeed, water stress leads to a decrease in photosynthesis and therefore in yields, but also to an increase in temperature, which itself can have an effect on yield. Therefore, to separate the thermal effect it is necessary to be able to control the water stress. This is the case with irrigated conditions and Figure 8b does not show a clear effect of heat stress under these conditions. The authors use an empirical model (eq 8) which is not at all suitable for separating the temperature and water effects or it should be demonstrated.

Your intuition is spot-on, and is exactly what the statistical methodology employed in this analysis achieves. That is, the coefficient on extreme heat is the partial effect of heat on yield, controlling for water stress; the coefficient on water stress is the partial effect of water stress on yield, controlling for heat. This is the advantage of a panel statistical model -- to be able to tease apart the two effects. It is true that there is some interaction between the two impact channels (as you mentioned,  water stress reduces photosynthesis and also raises plant temperature) and so these "heat" and "water stress" channels should be interpreted carefully. The temperature coefficient would represent the net of all effects raising surface temperature. We have added text to better explain this in the discussion in line 466 "As what we have shown, the cooling effect of irrigation lowers evaporative demand (PET) and thus indirectly contributes to lower water stress (higher ET/PET). In addition, water stress reduced photosynthesis and ET, resulting in higher plant temperature. Our disentangling methods do not account for the water stress and heat stress interaction effects, so these "heat" and "water stress" channels should be interpreted carefully. We note that our statistical model estimated temperature coefficient should be interpreted as the net of all effects raising surface temperature." .

Therefore i found that the conclusions (as the 65% and 35%, heat) cannot be supported by such methodology and probably the author give too much importance to the heat stress.

The 65% and 35% come from our statistical model, which (as explained above) is the state-of-the-science for statistical isolation of individual parameter impacts on yield. However, we understand the reviewer's intuition that moisture stress is the dominant factor. One of the surprising findings across the statistical crop-climate literature has been the persistent finding that heat itself matters, and not just through moisture. Our desire to explore this interesting finding is what prompted the analyses in this paper.

In fact this study does not refer to existing knowledge in ecosphysiology on heat stress on field crops which would have allowed a better understanding of the periods and impact of heat stress on yields. This is reflected in the choice of crop models, which is documented in a much too summary manner. Because of the strong link between water stress and temperature, on the one hand, and the pre-eminence of water stress on yields, it seems difficult to isolate the effect of heat stress on yield with a simple statistical analysis as is done in the paper. Moreover, the choice of explanatory variables aggregated over two parts of the cycle does not help to analyse phenomena that occur over short periods of time linked to climatic variability and the sensitivity of the yield to heat stress.

We agree that many processes over shorter time scales affect crops, including (eg) water stress raising plant temperature. We do not claim to be able to isolate the individual processes over the lifecycle of the crop, but instead seek to understand the net heat- vs water stress- driven impacts over the whole season. We agree that analyses complementary to ours (e.g., detailed phenological studies) are important for completing our understanding of the heat- and water- stress puzzle.

The main reason why we used the crop models is to test how well crop models had simulated the net irrigation benefits we identified in different phases of crop growth. We did not make a choice of the crop models but just used all available crop model results participating in GGCMI. It is indeed likely true that other crop models have a better representation in crop physiological response to climate stress and canopy temperature simulation. But here we find that only one of these readily-available models has a good performance in representing canopy temperature.

Because of the unsuitable approach to adress thermal stresses which would have been the true originality of this work, I do not recommend the publication of this article. Moreover, it has some formal defects:

We respectfully disagree. These statistical models are the state of the art for long panel analyses, and do in fact enable us to tease apart the various impact pathways. They may not be as detailed or process-based as the reviewer would prefer, but they are very useful for leveraging long time series data over large areas.

The authors could describe a little better the sources of performance data

We have detailed below answers to your specific questions about the data, and have added additional clarification to the manuscript accordingly.

L126 what is a MODIS sinusoidale projection

This is the projection system used by many MODIS products. This projection is pseudocylindrical equal-area projection displaying all parallels and the central meridian at the real scale. It uses a spherical projection ellipsoid and a WGS84 datum ellipsoid.

L141-144 : better describe how phenology is retrieved. Site observations in Figure 4c show that the phenology was not well characterized (gap of 20 days with VP this gap might have commented

The inconsistency between site observations and satellite derived crop phenology is likely due to the different spatial scales: Figure 4b shows the statistics of the whole state; while Figure 4c is only for the selected site. In addition, Figure 4c is presented here mainly to support our finding that irrigation extended the crop growing duration especially for the grain filling period but not to compare the absolute value of phenological stages derived from two different sources and scales. Actually, the satellite derived crop phenological stages have been validated against USDA statistics in the previous study for Nebraska state (Zhu et al., 2018). Please see the comparison below:

[Figure]

The two dashed lines in the figure define the region where the errors between satellite derived phenological stages and NASS statistics are less than 5 days.

Zhu, P.**,** Jin, Z.**,** Zhuang, Q., Ciais, P., Bernacchi, B., Wang, X., Makowski, D., Lobell, D. The important but weakening maize yield benefit of grain filling prolongation in the US Midwest. Glob Change Biol. 2018;00:1–13.

L176-180 : I guess that met data are obtained hourly, why using sine function (is the fact of using sine function has an impact on GDD and EDD

As its name (Daymet) indicated, the meteorological data are daily time-step, so we use this interpolation to better capture the sub-daily temperature stress. We clarified this in line 169 "we also obtained daily minimum and maximum surface air temperature (Tmin and Tmax) at 1-km resolution from Daymet version 3"

L301  not clear

We clarified this as "this suggested that irrigated and rainfed cropland were distinguishable based on satellite derived crop seasonality information".

L328-329: are irrigated and non irrigated using varieties. Probably not and we can expect that phenological characteristics might be different. This can explain shorter cycle with non irrigated crop.

Yes, irrigated and non irrigated corn fields probably use different varieties. So we also used Figure 4d-f to show how much the difference in growing duration can be partially explained by the LST differences.

L390-403 : in Agmip there several models that compute crop temperature (STICS for instance), wihy not using them. Are sure that LPJ-guess do no not compute crop temperature.

We do not deliberately make model selections but just used all models available in AgMIP Global Gridded Crop Model Intercomparison (GGCMI) project (Müller et al., 2019). Indeed, STICS simulates canopy temperature, but this model is not available in AgMIP GGCMI project.

Müller, C., Elliott, J., Kelly, D., Arneth, A., Balkovic, J., Ciais, P., Deryng, D., Folberth, C., Hoek, S., Izaurralde, R. C., Jones, C. D., Khabarov, N., Lawrence, P., Liu, W., Olin, S., Pugh, T. A. M., Reddy, A., Rosenzweig, C., Ruane, A. C., Sakurai, G., Schmid, E., Skalsky, R., Wang, X., de Wit, A. and Yang, H.: The Global Gridded Crop Model Intercomparison phase 1 simulation dataset, Sci. Data, 6(1), doi:10.1038/s41597-019-0023-8, 2019.

---

## Author Response (AR2)

**Reviewer 1**
The authors responded well to all reviewer comments and performed additional analyses as requested by the reviewers. The article itself is interesting and has the potential to further advance the scientific understanding of heat and drought effects on crop development and yield and corresponding effects of irrigation. I clearly see the merit of the work presented here. However, I also see a risk that the results will be misunderstood by many researchers and in particular by the media. The three reviewers and the authors agree that heat and drought effects cannot be completely separated because of a considerable interaction and that this interaction cannot be represented by the methodology used by the authors. What if the single effects of water and heat are small and the interaction explains most of the total effect? Consequently, I'm afraid that the central finding of 65% water alleviation effect and 35% heat alleviation effect of irrigation will be communicated in subsequent studies without pointing to the limitations and context of the present study. Clearly this should be avoided. The authors added an explanation to the discussion explaining these limitations but I still think that this will not help much to avoid misinterpretation. I suggest therefore to mention this aspect already in the abstract section of the article to highlight the relevance of this limitation. To make this very explicit: it is not my intention to lower the merit of the present work but to avoid misinterpretation! I suggest therefore to pack it in a way that one final sentence will be added to the abstract explaining that future research should make efforts to consider as well the interaction effects between heat and drought alleviation.

We thank the reviewer for the constructive suggestions, which have significantly improved our study. We have added text to the abstract suggesting that future research consider the interaction effects between heat and drought alleviation. "considering the potentially strong interaction between water and heat stress, future research on irrigation benefits should explore the interaction effects between heat and drought alleviation."

**Reviewer 2**
Compared to the previous version, the authors have consolidated the statistical analysis with an explicit consideration of collinearities. Nevertheless, I think that the results given for thermal stresses should be more discussed and the analysis of the results should be pushed further. The value of 35% could be compared with the results of the literature. For example, the study by Lobell et al. 2011 gives orders of magnitude of 1% yield loss per ℃ above 30 for well-watered crops and 1.7% per ℃ for poorly watered crops. In the study, the authors find a value of 1% per ℃ in the case of irrigated crops, which is the case where the thermal effect is well isolated from the hydric effect, as shown by the low uncertainty in the thermal sensitivity coefficient shown in Figure 7. Such results is comparable to that of Lobell and thus demonstrate the interest of the method used to aggregate the data of the study. Such a result is somewhat at odds with the strong effect of temperature found by the statistical analysis on yield variations between irrigated and non-irrigated systems, which I still believe suffers from uncertainties related to residual collinearities. A FIV of 2.2, while below the severe collinearity threshold of 5, still shows collinearities that give a 50% increased variance on the parameters. Could this explain a discrepancy between the results obtained under irrigated conditions where the effects of water and heat stress are well deconvoluted, but also the discrepancy with the Lobell 2011 study in case of water stressed crop 5(1.7% per ℃). I encourage the authors to really

consider this discussion, which does not questioned the interest of the paper which clearly demonstrates the need to take temperature into account in the operating models.

Thank you for this detailed and constructive review. We have made the suggested changes, as detailed in the point-by-point response below.

We appreciate the suggestion to compare our findings to Lobell et al 2011. We want to clarify an important point -- the estimation reported by Lobell et al. 2011 is based on degree days (GDD$_{30+}$, which measures sum of temperature exposure above 30 ℃) and not the mean temperature. So the 1% yield loss reported is per degree day, not per ℃. However, Fig 2b in Lobell et al. 2011 shows the yield sensitivity of 1 ℃ warming. We can see the poorly watered maize shows high sensitivity (<-10% per ℃) to warming stress based on the estimated yield sensitivity.

We have added a sentence in line 382-384 to show the similar findings of the previous studies on the weakened temperature sensitivity of yield due to irrigation application. "We found that irrigation not only lowered water and high temperature stress, but also made yield less sensitive to water and high temperature stress (Figure 8a-c), consistent with previous studies (Troy et al., 2015; Tack et al., 2017). For example, field data across Africa suggests that better water management can reduce yield loss due to heat stress from -1.7% per degree days to -1% per degree days (Lobell et al., 2011). "

From a statistical point of view, it would be good to define what the uncertainties reported in the paper correspond to (error bar on figure 7, % given on the contribution of water and thermal effects on page 13.
Thanks for your suggestion. We have added details about what those uncertainties represent for both Figure 7 and the contributions of water and thermal effects reported on page 13.

It would be good to give the VIF for the adjustment of the parameters of equation 7. It is likely to be high in the case of rainfed crops.
We added the VIF in Line 369-371 for equation 7 for both irrigated and rainfed maize yield. Then we find the VIF for equation 7 is still not very high with 2.8 and 3.6 for irrigated and rainfed maize yield, respectively. We added this in Line 369-371 "we quantified the variance inflation factor (VIF) in the model; this was found to be well below standard thresholds, with a value of 2.8 and 3.6 for irrigated and rainfed maize yield, respectively"

L200 : I disagree. In figure 5 of supplemental, I saw rather an overestimation of the Modis PET. The MODIS PET data were always at the upper limit of the observation data. Taking time average will exhibit bias (~15-25% from a visual impression). This might explain the low ET/PET ratio obtained with irrigated fields (I expected values closer to 1 and I am surprised that some irrigated fields led to ET/PET <0.6) - What is the consequence of such overestimation on the study
Indeed, we confirmed that across all site-year samples, MODIS PET is on average 17% higher than the observation data, which partially explained the relatively lower ET/PET in irrigated fields. However, the main reason for the lower than expected ET/PET is because current irrigation systems cannot ensure crops are fully irrigated

during the growing season as it is not precision irrigation and tends to be run somewhat infrequently due to the lumpy expenditure.

Obviously, overestimation in PET will impact the water stress and heat stress alleviation contribution. We simulate the effect of overestimated PET on our attribution analysis by adding a 17% positive offset to all PET samples. We find the mean of ΔEP/PET between irrigated and rainfed maize changed from 0.088 to 0.075 as the following figure shows.

[Figure]

Then we apply the equation (9) to ΔET/PET samples with 17% positive offset and ΔLST to estimate the water stress and heat stress alleviation contribution. We find the yield improvement due to water and high temperature stress alleviation is 61±10% and 39±5.7%, which is very close to the original estimation that water and high temperature stress alleviation contributed to 65±10% and 35±5.3% yield improvement. With this simulation, it is safe to say that overestimation in PET has a small effect on the attribution analysis and will not change our overall findings.

L340 : I would add here the variety effect.
Thanks for your suggestion. We have added this potential effect in Line 340: "(3) variety differences between irrigated and rainfed maize"

L340-342 not really useful, consider to remove such statement
We followed the suggestion and removed the two lines and updated the figure accordingly.

360 : is 2% really significant ? (further a -2% was found not significant (375-376).
Here the 2% difference is not significant. We have added the significance level for

this comparison. The -2% temperature sensitivity estimation in L375-376 is significant. We added signs in Figure 7 to represent the significance of temperature sensitivity estimation.

[Figure]

**Figure 7:** Temperature sensitivity of yield and yield components (GSL, HI and BGR) for irrigated and rainfed maize areas. The error bars represent the 95% confidence interval of estimated temperature sensitivity. ** indicates a significant estimation of temperature sensitivity with p<0.01 while * indicates significance with p<0.05.

429-431 : I am not sure that scale of study was really helpful to farmers. Heat and drought tolerance are likely analysed in selection plan. I would removed such statement
We followed the suggestion and have removed these statements.

447-452 : I disagree, several models already take crop temperature into account as well as the interaction between heat and water stresses (CLM, STICS, CERES?, ). It would have been interesting to analyse in the discussion how heat stress are addressed in such models and if the order of magnitude found if the statistical analysis are retrieved (for instance by making numerical experiments).

Indeed, there are some crop models (like STICS and APSIM) already taking canopy temperature into account when addressing heat stress. We have added some discussion about how canopy temperature stresses are estimated in these models in line 456-468. "Relatedly, recent studies compared heat stress representation in crops models which explicitly simulate canopy temperature (Webber et al., 2017). For example, STICS estimates canopy temperature using canopy energy balances which account for net radiation, soil heat flux, evapotranspiration and aerodynamic resistance (Brisson et al., 2003). In APSIM, canopy temperature is taken as 6 ℃ higher than air temperature when the crop is fully stressed and 6 ℃ cooler than air temperature when the crop is fully transpiring. Between these limits, the basis of the

expression for canopy temperature is the relationship between temperature difference (canopy temperature minus air temperature) and the ratio of actual and potential evapotranspiration (Webber et al., 2017). This model comparison study suggests that models using canopy temperature to account for heat stress effects indeed outperform those models depending on air temperature but the model comparison also identified a wide range for the simulated canopy temperature in current crop models. ''

For the crop models we used in AgMIP project, CLM-crop is the only crop model taking canopy temperature into account. We also do not find any numerical model experiments that did such analysis like ours to disentangle the heat and water stress effects. Therefore, we suggest crop model communities pay more attention to this issue and set up more model experiments by including crop models explicitly considering canopy energy balance to better address heat stress and water stress alleviation due to irrigation and thus better simulate crop yields with human management practices.